# Three delays model applied to pediatric injury care seeking in Northern Tanzania: A mixed methods study

Elizabeth M. Keating[1]*, Francis Sakita[2,3], Blandina T. Mmbaga[3,4,5], Ismail Amiri[5], Getrude Nkini[5], Sharla Rent[6], Nora Fino[7], Bryan Young[1], Catherine A. Staton[8,9,10‡], Melissa H. Watt[10,11‡]

1 Department of Pediatrics, University of Utah, Salt Lake City, Utah, United States of America, 2 Emergency Medical Department, Kilimanjaro Christian Medical Centre, Moshi, Tanzania, 3 Kilimanjaro Christian Medical University College, Moshi, Tanzania, 4 Kilimanjaro Clinical Research Institute, Moshi, Tanzania, 5 Duke-KCMC Collaboration, Kilimanjaro Christian Medical Centre, Moshi, Tanzania, 6 Department of Pediatrics, Duke University Medical Center, Durham, North Carolina, United States of America, 7 Department of Internal Medicine, University of Utah, Salt Lake City, Utah, United States of America, 8 Department of Emergency Medicine, Duke University Medical Center, Durham, North Carolina, United States of America, 9 Global Emergency Medicine Innovation and Implementation (GEMINI) Research Center, Duke University Medical Center, Durham, North Carolina, United States of America, 10 Duke Global Health Institute, Duke University, Durham, North Carolina, United States of America, 11 Department of Population Health Sciences, University of Utah, Salt Lake City, Utah, United States of America

‡ These authors contributed equally to this work and are co-senior authors
* Elizabeth.Keating@hsc.utah.edu

## Abstract

Pediatric injuries are a leading cause of morbidity and mortality in low-and middle-income countries. Timely presentation to care is key for favorable outcomes. The goal of this study was to identify and examine delays that children experience between injury and receiving definitive care at a zonal referral hospital in Northern Tanzania. Between November 2020 and October 2021, we enrolled 348 pediatric trauma patients, collecting quantitative data on referral and timing information. In-depth interviews (IDIs) to explain and explore delays to care were completed with a sub-set of 30 family members. Data were analyzed according to the Three Delays Model. 81.0% (n = 290) of pediatric injury patients sought care at an intermediary facility before reaching the referral hospital. Time from injury to presentation at the referral hospital was 10.2 hours [IQR 4.8, 26.5] if patients presented first to clinics, 8.0 hours [IQR 3.9, 40.0] if patients presented first to district/regional hospitals, and 1.4 hours [IQR 0.7, 3.5] if patients presented directly to the referral hospital. In-hospital mortality was 8.2% (n = 30); 86.7% (n = 26) of these children sought care at an intermediary facility prior to reaching the referral hospital. IDIs revealed themes related to each delay. For decision to seek care (Delay 1), delays included emergency recognition, applying first aid, and anticipated challenges. For reaching definitive care (Delay 2), delays included caregiver rationale for using intermediary facilities, the complex referral system, logistical challenges, and intermediary facility delays. For receiving definitive care (Delay 3), wait time and delays due to treatment cost existed at the referral hospital. Factors throughout the healthcare system contribute to delays in receipt of definitive care for pediatric injuries. To minimize delays and improve patient outcomes, interventions are needed to improve caregiver and healthcare

**Data Availability Statement:** The data for this manuscript is covered under a data sharing agreement, and thus we are unable to openly share these data. For data access requests, please

contact our non-author KCMC representative Gwamaka William at gwamakawilliam14@gmail.com.

**Funding:** EMK was supported in this work by the Fogarty International Center of the National Institutes of Health (D43 TW009337). The content is solely the responsibility of the authors and does not necessarily represent the official views of the National Institutes of Health. The funders had no role in study design, data collection and analysis, decision to publish, or preparation of the manuscript.

**Competing interests:** The authors have declared that no competing interests exist.

worker education, streamline the current trauma healthcare system, and improve quality of care in the hospital setting.

## Introduction

Injuries are a leading cause of death in children worldwide [1]. More than 95% of pediatric injury-related deaths occur in low-and middle-income countries (LMICs) [1] with children in sub-Saharan Africa even more disproportionately affected [2]. There are many contributors to this problem, including inadequate road infrastructure and traffic laws, and lack of specialized services such as Emergency Medical Systems and healthcare [2]. Much of the data on pediatric injuries in sub-Saharan Africa is epidemiological and from registries [3–9]. There is a paucity of data on strategies to improve the care and outcomes of injured children in LMICs, and thus an important knowledge gap exists [10,11].

When a child is injured there are many steps to receiving appropriate care, starting with the recognition of the emergency by the caregiver and continuing through to the care provided in a healthcare facility. There are many places in this pathway to care where delays can exist. Timely presentation and receipt of care is key for favorable outcomes in injured children, especially in those with severe injuries, and thus it is important to describe delays to care in order to understand and address them [12,13].

The Three Delays Model was developed to understand delays during obstetrical emergencies that contribute to maternal mortality [14,15]. The model describes three important delays to care including: (1) the decision to seek care, (2) reaching definitive care, and (3) receiving definitive care. The model provides a useful framework to examine factors that influence the timely receipt of care at each stage. The Three Delays Model has been applied to understand the delays to care in other settings, including additional areas of maternal health, neonatal mortality, pediatric cancer, and emergency care seeking [16–23]. In each of these settings, the relative contribution of each of the different delays varied [16]. The Three Delays Model is a useful tool to map patient experiences and identify site-specific delays.

Given the limited data on pediatric injuries and the importance of injured children receiving timely care, this exploratory study aimed to identify and investigate delays that children experience between injury and receiving definitive care at a zonal referral hospital in Northern Tanzania. To accomplish this, we used the Three Delays Model framework in order to understand the presence and context of each delay. The findings can inform locally relevant, targeted interventions to shorten delays to care and subsequently reduce morbidity and mortality from pediatric injuries in this region.

## Materials and methods

### Study design

This is a mixed methods study involving quantitative registry data and qualitative semi-structured in-depth interviews (IDIs) [24]. To understand the delays in care seeking that occurred after injury in pediatric patients, we used a mixed-methods explanatory design analysis. We first used quantitative registry data to identify timing delays to care in pediatric injury patients. We then explained and explored these delays using qualitative IDI data utilizing the Three Delays Model framework [14]. This study and all procedures received ethical approval from the institutional review boards at the Tanzanian National Institute for Medical Research, Kilimanjaro Christian Medical Centre, and the University of Utah.

### Inclusivity in global research

Additional information regarding the ethical, cultural, and scientific considerations specific to inclusivity in global research is included in the Supporting Information (S1 Checklist).

### Study setting

The study took place at Kilimanjaro Christian Medical Centre (KCMC), a zonal referral hospital for the northern regions of Tanzania (Arusha, Dodoma, Kilimanjaro, Singida, and Tanga). KCMC is one of four zonal referral hospitals in Tanzania, and has a catchment population of 12,000,000 persons. KCMC is in the Kilimanjaro region of Northern Tanzania, and serves as a referral site for children with injuries requiring specialty investigations, imaging, and treatment. The Emergency Medical Department (EMD) at KCMC sees approximately 1400–1700 pediatric patients per year. KCMC has general surgeons and orthopedic surgeons who treat trauma patients, including children. In addition, KCMC has pediatricians that assist in the care of pediatric trauma patients. There is a specialized burn center at KCMC. There is no formal emergency medical system in Tanzania.

### Participants

The KCMC pediatric trauma registry was launched in November 2020 in order to define areas for quality improvement in the care of pediatric trauma patients. Patients were enrolled in the registry on presentation to the KCMC EMD if they were less than 18 years old and presented with an injury. Injuries, as defined by the World Health Organization, are caused by acute exposure to physical agents such as mechanical energy, heat, electricity, chemicals, and ionizing radiation interacting with the body in amounts that exceed the threshold of human tolerance [25]. This registry enrolled patients who were injured with fractures, burns, lacerations, traumatic brain injuries, ingestions/poisonings, animal envenomations, road traffic injuries, falls, drownings, penetrating trauma, non-accidental trauma, and others. Registry inclusion criteria included pediatric patients seeking care for any injury that occurred in the last month who survived to evaluation in the EMD. Patients were not included in the registry if they were presenting for injury follow-up care.

Caregivers were eligible for semi-structured IDIs if they were the caregiver (such as mother, father, other relative) of a pediatric injured patient who was admitted to KCMC from the EMD, fluent in Kiswahili, and could provide informed consent. A convenience sample of 30 family members were recruited by a research assistant face-to-face and were invited to participate in an interview. Attention was given to interviewing caregivers who would be good informants. In order to adequately represent our patient population, we attempted to interview caregivers of patients with a variety of ages, injury mechanisms, and injury types. The number of participants recruited was pre-set and informed by team-based discussions of when thematic saturation was likely to occur.

### Quantitative procedures

The pediatric trauma registry prospectively enrolled all patients less than 18 years of age presenting to the KCMC EMD for treatment of an injury. All registry data were recorded on tablets in REDCap [26], and quality of all entries was reviewed by the Principal Investigator (EMK). For the purposes of this analysis, variables were extracted from the REDCap database regarding: demographics, mechanism of injury, time of injury, mechanism of transport, first health facility treated at, time of first care, time of arrival to KCMC, and patient outcomes including in-hospital mortality. We grouped first sites of care as: (1) clinics (local dispensaries

or health centers), (2) district or regional hospitals, and (3) KCMC. Patients with missing outcomes were excluded.

## Quantitative data analysis

To characterize care seeking delays of injured children, descriptive statistics summarized demographics, timing of care, and referrals. Continuous data were represented as means with standard deviation or medians with interquartile ranges (IQR). We compared survivors and non-survivors across key outcomes demographics and treatment characteristics using analysis of variance (ANOVA), Fisher's exact test, or Chi-squared tests as appropriate. All analyses were performed in SAS (Version 9.4).

## Qualitative procedures

Semi-structured IDIs were completed with 30 family members of injured patients. The interview guide was developed in order to understand delays that occurred along the care continuum from injury to receiving definitive care. Family member participants were asked: when the injury occurred, when the decision was made to seek care, if there was a delay in care seeking and why, where they went to receive care, why they chose to go there first, why they were transferred to KCMC, if there was a delay between referral and transfer and why, and how long they waited for care in the KCMC EMD. The guide was translated from English to Kiswahili by two native speakers, checked for accuracy in translation meetings by a team of native speakers, and pilot tested by the research team.

IDIs were conducted between November 2020 and October 2021. All IDIs were completed in Kiswahili by two trained bilingual Tanzanian research assistants (IA and GN) who had extensive experience conducting qualitative interviews. The research assistants completed two weeks of training that included qualitative interview strategies and gaining familiarity with the study protocol and qualitative interview guides. IDIs were conducted in quiet, private locations within KCMC near the end of the patients' hospital stay. Each IDI took approximately one hour. Participants were provided a copy of the consent form, which was read aloud by research assistants. Participants provided their written informed consent prior to completing the IDIs, and participants unable to write provided a thumbprint in front of a witness. Participants received 10,000 Tanzanian shillings (approximately 4 US Dollars) for their time. IDIs were audio recorded for verbatim transcription. Interview audio recordings were transcribed and translated to English by the two bilingual Tanzanian research assistants.

## Qualitative data analysis

De-identified data were analyzed through a team-based, thematic approach informed by applied thematic analysis with qualitative memo writing [27,28]. After multiple readings of the IDI transcripts, a memo summarizing each transcript was written by investigator BY after training by EMK and MHW. Memos followed an established template of a priori domains, informed by the interview guide, to extract and synthesize the text's core meaning related to the research questions and to extract representative quotes. The memo-writing process identified emerging themes and informed the development of preliminary codebooks. Each memo was on average two single-spaced pages long and reviewed by EMK to ensure that the interview content was being interpreted appropriately. After the memos were reviewed, investigators BY, EMK, MHW, and SR met to reach team consensus on the emerging themes and finalize the codebook. The memos were then coded in Dedoose software, and the codebook was continually adapted to reflect new or emerging themes as coding progressed. All memos were double-coded by BY and SR, with intercoder discrepancy discussed amongst the research

team and resolved by consensus, with EMK determining final application of codes if disagreements remained.

IDIs were analyzed using the Three Delays Model framework in order to understand the presence and context of each delay. We define Delay 1, or the decision to seek care, as delays that occur after the injury up until the decision to seek care has been made. We define Delay 2, or reaching definitive care, as delays that occur after the decision to seek care has been made up until reaching definitive care at KCMC. We define Delay 3, or receiving definitive care at KCMC, as delays that occur after reaching KCMC until receiving definitive care at KCMC.

## Results

### Quantitative

A total of 365 patients were enrolled in the pediatric trauma registry from November 2020 to October 2021. Demographic characteristics of the patients are described in Table 1. Overall, 81.0% (290) of pediatric injury patients at KCMC were referred from an intermediary health-care facility. Only 22.62% of our patients had health insurance, while 76.32% paid with personal or family funds. For patients first treated at clinics, the median time from injury to arrival at KCMC was 10.2 hours [IQR 4.8, 26.5] compared to 1.4 hours [IQR 0.7, 3.5] in patients who went straight to KCMC (Fig 1). In patients first treated at district or regional hospitals, the median time from injury to arrival at KCMC was 8.0 hours [IQR 3.9, 40.0]. Overall, 30 patients (8.2%) died at KCMC; of these, 26 (86.7%) had gone to an intermediary facility before reaching KCMC (Fig 2).

### Qualitative

The IDIs with caregivers revealed themes around each of the three delays (Table 2).

### Delay 1: Decision to seek care

Four themes emerged during delay 1: recognizing emergency, home first aid and traditional medicine, anticipated challenges to seeking care, and fear of seeking care.

**Recognizing emergency.** Most caregivers recognized the emergency and immediately sought care, often citing injury severity as a driving factor. Caregivers that failed to recognize the emergency, and delayed seeking care, described waiting to seek care until it appeared that their child was worsening.

> *I think they thought it was not severe. . .the next day when they asked him to wake up. . .he told the grandmother that all his body parts were not functioning. That is when I went to pick him up to go to the hospital. (Father of a 9-year-old male with a fall causing a TBI. First care was received at a dispensary and the patient arrived at KCMC 4 days after injury. IDI 15)*

**Home first aid and traditional medicine.** The application of home first aid prior to seeking clinical care was commonly seen in the setting of burn injuries, with caregivers applying cold water, honey, or raw eggs to burn wounds. For injuries other than burns, home first aid included massage of fractured wounds and applying basic bandages.

> *What I did first, I took honey to cover his wound then I prepared myself to take him to the hospital. (Mother of a 19-month-old male with a burn injury. First care was received at home and the patient arrived at KCMC 43 minutes after injury. IDI 23)*

**Table 1. Demographics.**

| Characteristic | Overall | In-Hospital Mortality | Discharged | P value |
|---|---|---|---|---|
| | | (n = 30) | (n = 335) | |
| Sex, N (%) | | | | |
| Male | 240 (65.8%) | 18 (60.0%) | 222 (66.3%) | 0.488 |
| Female | 125 (34.2%) | 12 (40.0%) | 113 (33.7%) | |
| Age, years, mean (SD) | 7.4 (5.0) | 6.3 (5.0) | 7.5 (5.0) | 0.233 |
| Age Group, N (%) | | | | |
| 1. Infant (0–1 year) | 43 (11.8%) | 7 (23.3%) | 36 (10.7%) | 0.313 |
| 2. Toddler (2–3 years) | 58 (15.9%) | 5 (16.7%) | 53 (15.8%) | |
| 3. Preschool (4–5 years) | 51 (14.0%) | 2 (6.7%) | 49 (14.6%) | |
| 4. Child (6–11 years) | 126 (34.5%) | 10 (33.3%) | 116 (34.6%) | |
| 5. Teen (12–17 years) | 87 (23.9%) | 6 (20.0%) | 81 (24.1%) | |
| Where patient lives, N (%) | | | | |
| Moshi Urban District | 106 (29.0%) | 3 (10.0%) | 103 (30.7%) | 0.028 |
| Moshi Rural District | 113 (31.0%) | 10 (33.3%) | 103 (30.7%) | |
| Other (rural surrounding areas) | 146 (40.0%) | 17 (56.7%) | 129 (38.5%) | |
| Mechanism of injury*, N (%) | | | | |
| Fall | 137 (37.5%) | 7 (23.3%) | 130 (38.8%) | 0.094 |
| Burn | 40 (11.0%) | 13 (43.3%) | 27 (8.1%) | <0.001 |
| Road Traffic Injury | 125 (34.2%) | 7 (23.3%) | 118 (35.2%) | 0.189 |
| Other | 190 (52.1%) | 10 (33.3%) | 180 (53.7%) | 0.032 |
| Mechanism of Transport to KCMC | | | | |
| Ambulance from other hospital | 202 (55.3%) | 24 (80.0%) | 178 (53.1%) | <0.001 |
| Private car | 56 (15.3%) | 4 (13.3%) | 52 (15.5%) | |
| Hired transportation (bajaji, boda boda, taxi) | 92 (25.2%) | 0 (0.0%) | 92 (27.5%) | |
| Walking | 2 (0.5%) | 0 (0.0%) | 2 (0.6%) | |
| Other (bus, police car, office car, etc) | 13 (3.5%) | 2 (6.7%) | 11 (3.3%) | |
| First health facility treated at, N (%) | | | | |
| KCMC | 68 (19.0%) | 2 (7.1%) | 66 (20.0%) | 0.229 |
| Clinic | 97 (27.1%) | 8 (28.6%) | 89 (27.0%) | |
| District/regional hospital | 193 (53.9%) | 18 (64.3%) | 175 (53.0%) | |

*Children could have more than one mechanism of injury.

The use of traditional healers was also mentioned as a delay. One caregiver described pressure from her family to take her child to a traditional healer, in part due to her inability to afford transportation to the local allopathic health center.

**Anticipated challenges to seeking care–place of injury and time of injury.** Many caregivers described anticipated challenges to seeking care for their injured child. One anticipated challenge was the place of injury, as some caregivers described living far from healthcare in a very rural area. In addition, many caregivers anticipated the challenge of the time of day the injury occurred. These caregivers noted that their child's injury occurred at night, and thus they could not seek immediate care given the lack of transport, lack of ability to collect money from family and friends, or lack of ability to walk to the health center safely.

*I don't have the ability to find money at night hours because it is risky for a woman to walk at night with a baby on their back. (Mother of a 4-year-old female with an eyelid laceration.*

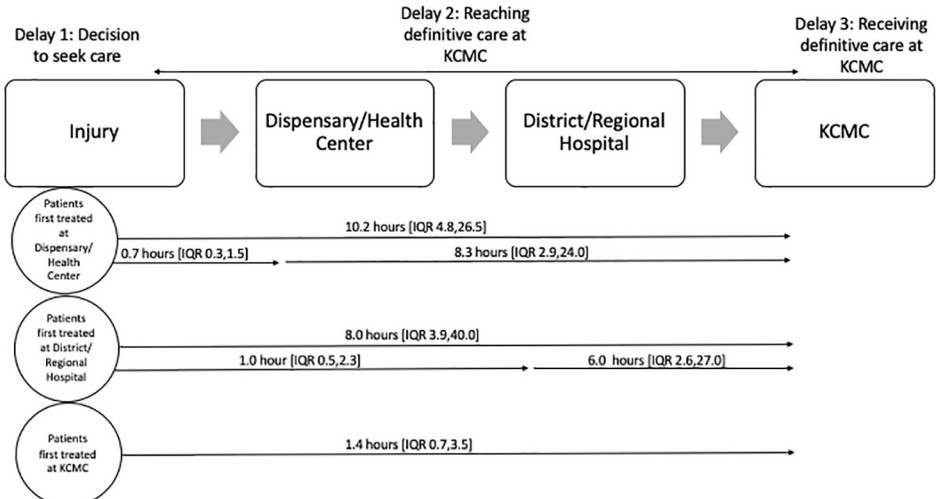

**Fig 1. Three Delays Model [14] as applied to pediatric injuries in Northern Tanzania.** This figure shows the usual pathway from injury to definitive care at KCMC, in which patients often seek care first at a clinic (local dispenary or health centre), and then are referred through the healthcare system in a stepwise fashion, first to a district hospital, then to a regional hospital, and finally to a zonal referral hospital like KCMC. Although this is the usual pathway to care in this healthcare system, there are instances where patients may skip steps along this pathway, and go from a clinic to KCMC directly, for example.

*First care was received at home and then at a dispensary, and the patient arrived at KCMC 1 day after injury. IDI 20)*

**Fear of seeking care.** Some caregivers mentioned that there was a delay in taking the child to a hospital due to fear of getting in trouble with the police. One caregiver described a situation in which bystanders did not know what to do and did not seek care while the child laid unconscious for 1 hour 45 minutes with a road traffic injury.

*People feared to touch her when she was unconscious; they thought she had died already, so they feared what they would say to the police when they came. (Mother of a 7-year-old female with a clavicle fracture and TBI. First care was received at a Regional hospital and the patient arrived at KCMC 3 hours 25 minutes after injury. IDI 28)*

## Delay 2: Reaching definitive care

Four themes emerged related to reaching definitive care for a pediatric injury: caregiver rationale for choosing intermediary facilities, the referral system, logistical challenges, and intermediary facility delays.

**Caregiver rationale for choosing intermediary facilities.** When asked why they did not seek care at KCMC first, most caregivers cited the need for immediate treatment nearby, preferring to seek care at the health facility closest to them.

*We wanted our child to get quick care and first aid then ready to go [to] other hospital. We also considered the distance, it (the clinic) was nearby. (Mother of a 6-year-old female with a burn injury. First care was received at home and then at a dispensary, and the patient arrived at KCMC 14 days after injury. IDI 25)*

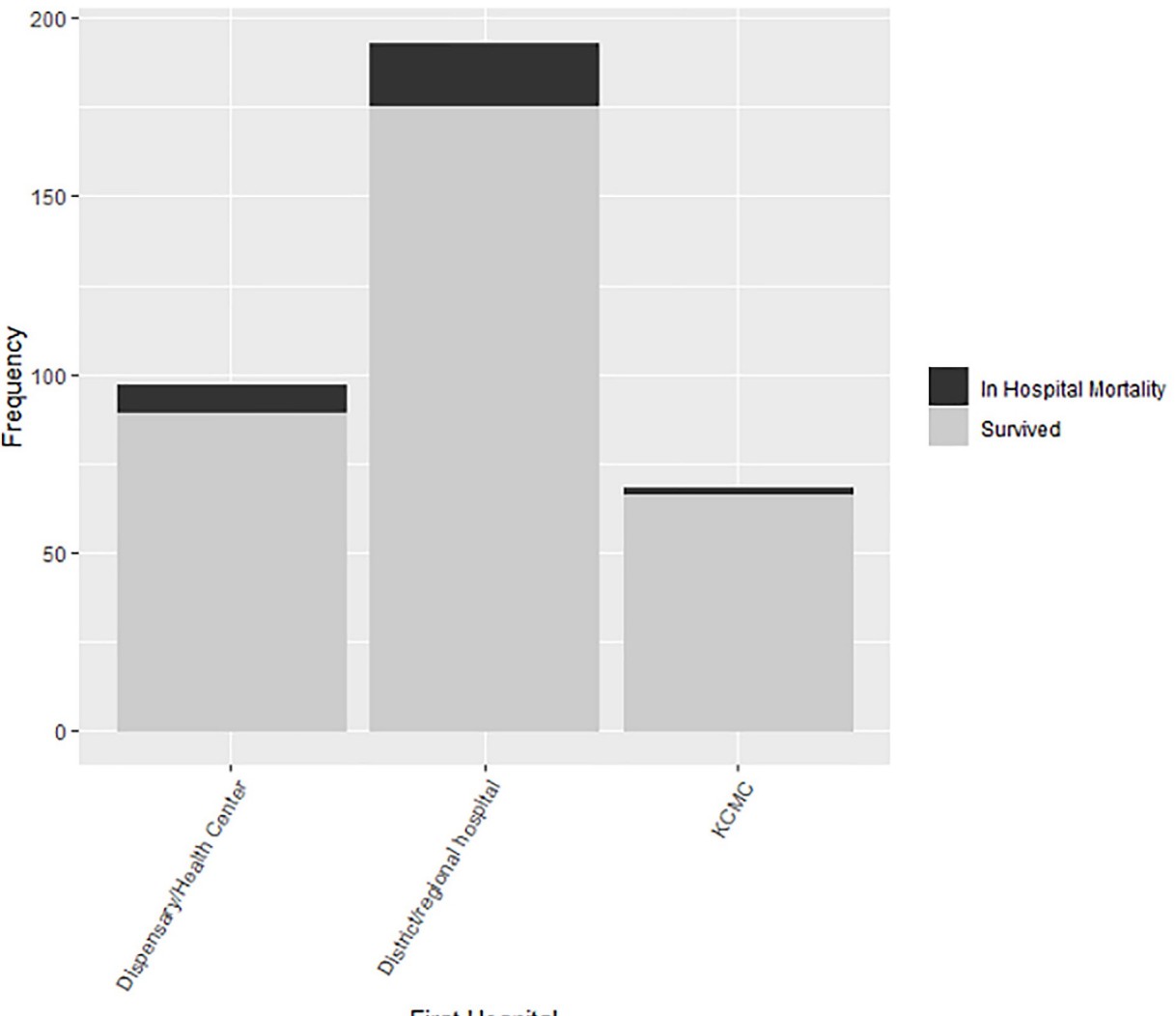

**Fig 2. Frequency of injured pediatric patients by first health facility.** This figure shows the frequency of injured pediatric patients by first health facility where they received treatment. It represents both those that survived (grey) and those that died (black).

**Table 2. Themes relating to the three delays for pediatric injury patients.**

| | *Emerging Themes* |
|---|---|
| *Delay 1*: Decision to seek care | Recognizing emergency |
| | Home first aid and traditional medicine |
| | Anticipated challenges to seeking care–place/time of injury and money |
| | Fear of seeking care |
| *Delay 2*: Reaching definitive care at KCMC | Caregiver rationale for choosing intermediary facilities–immediate treatment and referral system |
| | Logistical challenges–money and transportation |
| | Intermediary facility delays |
| *Delay 3*: Receiving definitive care at KCMC | Wait time |
| | Delays due to treatment cost |

Further, one caregiver noted that she chose a nearby private facility for first care due to the fact that the injury occurred late and she thought the private hospital would have services available even at night.

**Referral system.** Caregivers often cited the established referral system in their decision to seek care initially at intermediary facilities as opposed to KCMC. At each level of care, injured patients must receive a referral letter before transfer to the next level of care.

*It is a must to start at small health center. At our home you can't go straight to [district hospital]. So, you have to start at small hospital [where] they will give you a letter to go to big hospital, and the big hospital will either give you referral to KCMC or other hospital, but letter should come from there first. (Mother of a 4-year-old female with an eyelid laceration. First care was received at home and then at a dispensary, and the patient arrived at KCMC 1 day after injury. IDI 20)*

From interviews, we learned that patients would often visit 1–4 intermediary health facilities prior to arriving at KCMC, sometimes just stopping at the intermediary facility for a referral letter and not receiving care.

*They said they are not able to help my child so they gave us a referral letter to [hospital]. I went at [hospital] where I got nice reception and. . .they also said they are not able to treat her, so I have to go to KCMC. (Mother of a 4-year-old female with an eyelid laceration. First care was received at home and then at a dispensary, and the patient arrived at KCMC 1 day after injury. IDI 20)*

**Logistical challenges–money and transportation.** Various logistical challenges including money resulted in patients getting stuck in the referral system and prevented them from leaving intermediary facilities. Caregivers described the inability to afford hospital bills or transport at the initial health facility to the hospital, or having to return home to acquire funds rather than seeking appropriate care at KCMC when referred. Two caregivers described not having the money to pay for the bills at the initial hospital, but the hospital would not release the patients until the caregiver paid the bill.

*I asked them to take only fifty shillings (0.02 US Dollars) at first because I had no more money at that time then I would add more money later on, but they refused and told me, "your child will die here. Make sure you pay all the bills before leaving this hospital." So I had to find more money quickly. As I was finding money to pay, they called me again and told me, "the condition of your son is getting worse and we won't let you go anywhere without that money.". . .I had found already two hundred thousand so I sent [it] and they quickly switched the ambulance and brought [patient] to KCMC. (Mother of a 13-year-old male with a TBI, tension pneumothorax, and pulmonary hemorrhage. First care was received at a health center, and the patient arrived at KCMC 6 hours 45 minutes after injury. IDI 9)*

Beyond financial constraints, transportation between health centers was a common cause of delays. Many caregivers described having to walk hours from their remote village to obtain care for their child as transport was hard to find. For transport between health centers, families often had to wait for hours for an ambulance to become available or to allow for multiple patients to be transported simultaneously.

*At our village transportation is not easy especially for this case we had to look for a private car to come to the hospital. (Mother of a 10-year-old female with a burn injury. First care received at home and then at a health center, and the patient arrived at KCMC 4 days after injury. IDI 13)*

**Intermediary facility delays.** Intermediary facility delays at the dispensary/health center or district regional/hospital level included misdiagnosis or sending patients home rather than referring them to KCMC. In addition, it was common for patients to have to wait for a doctor to arrive in order to see the patient or sign off on transfer.

*When we arrived there, we didn't find [the] doctor in charge but we found a nurse who checked him and said "this leg has a fracture." So, we asked her to give their ambulance so that we can take [patient] to [the hospital] but the nurse said the doctor in charge is the one to give permission to use this car. Therefore, we are not able to sign to give you the car. (Aunt of a 10-year-old male with a femur fracture. First care received at a health center, and the patient arrived at KCMC 5 days after injury. IDI 17)*

Another common cause of delay was admission to the district/regional hospital first, even if what was needed (such as an x-ray) was not available in that hospital. This resulted in delays from hours to a week. Further, in some cases there was a failure to recognize that the patient was worsening or that the wound was severe enough to require care at KCMC. A number of caregivers described requesting a referral to KCMC but the providers at the district/regional hospital refusing.

*After the accident we took [patient] to [hospital]. But after two days I did not see any improvements and the wound was changing color with pus on it. When I asked the nurse to tell me what ways they are using to treat [patient] she told me we are only giving her first aid here. So I asked her why don't you give us a referral to another hospital then because we are wasting more time and money here. This is the second day and she might even die. She then went to discuss with her fellow [staff] and she called me and told me they will refer me [to KCMC]. (Caregiver of a 7-year-old female with a burn injury. First care received at district/regional hospital, and the patient arrived at KCMC 4 days after injury. IDI 26)*

Other intermediary facility delays included delays in preparing and printing the referral letter, and a misunderstanding about lack of services on the weekend. One caregiver described that their child was seen on a Friday and given a transfer letter to KCMC, but told to come back on Monday saying there were no orthopedic services at KCMC over the weekend, which is not true.

## Delay 3: Receiving definitive care at KCMC

Two themes did emerge regarding receiving definitive care at KCMC: wait time and delays due to treatment cost.

**Wait time.** The majority of caregivers commented that they did not experience long wait times after arrival to the EMD. Only three caregivers described long wait times ($> 2$ hours) after arrival prior to being seen by a doctor.

*It was a challenge to wait for that long at emergency. I can't say they don't care, but there is a way they don't care much, because it was not like all doctors were busy. No, you can see them talking, but since there is not anyone you know, so when you say like I am not yet seen, they*

*tell you to wait. So, if the condition of the baby is serious, there is a possibility of losing his/her life while at emergency and they will tell you it is because of the accident. (Mother of a 7-year-old female with a clavicle fracture and TBI. First care was received at a regional hospital, and the patient arrived at KCMC 3 hours 25 minutes after injury. IDI 28)*

**Delays due to treatment cost.** The most common delay in care and treatment mentioned was the inability to pay for testing, as many caregivers had spent most of their money at intermediary facilities described in delay 2. Some caregivers described needing to get an exemption letter from the Social Welfare Officer at the EMD prior to receiving treatment. Two caregivers described needing to return home to the village to collect money to pay for the child's x-rays and other testing, which caused delays ranging from hours to two-three weeks. In addition, after admission sometimes tests are delayed due to inability to pay.

*It took almost two to three weeks because I went to search for money and after two weeks when I got money. . .they did an x-ray. (Father of a 15-year-old male with a femur fracture and osteomyelitis. First care was received at a regional hospital, and the patient arrived at KCMC 2 months after original injury and 3 days after re-injury. IDI 27)*

## Discussion

This is the first study in which the Three Delays Model framework was applied to pediatric injury care delays in a LMIC. In this study, quantitative registry data showed that the majority of pediatric injury patients seen at KCMC were referred from an intermediary healthcare facility, and these patients presented much later than patients who presented to KCMC for first care. Many patients visited more than one intermediary facility prior to care at KCMC and experienced significant delays in doing so. In addition, there was a higher, though non-significant, mortality at KCMC for patients that initially presented to intermediary facilities as opposed to those who presented directly to KCMC. In order to explain and explore these findings, we conducted qualitative IDIs applying the Three Delays Model framework to pediatric injury care seeking in this setting and identified barriers that contribute to delays along the care continuum. These barriers could be addressed with interventions including caregiver education, streamlining the current trauma healthcare referral system, and continuing to improve quality of care for pediatric injury patients in the hospital setting.

Our study findings identified delays at the community level that are likely due to lack of caregiver knowledge surrounding what to do when a child is injured. These findings are supported by other studies that applied the Three Delays Model to access to care for pediatric surgery patients, pediatric patients, and emergent patients in LMICs and found that care-seeking delays were common [16,29,30]. In order to combat these delays, caregiver education on immediate first aid, recognizing the emergency, and when to seek care are needed [29]. One study in rural Ghanaian communities found that both appropriate and inappropriate first aid was applied by caregivers at home, and highlighted the need to increase context-appropriate, community-based first aid training programs for communities in LMICs [31].

The most common delays found in our study occurred in reaching definitive care at KCMC (Delay 2), and included the complex referral system. Along the care continuum in Northern Tanzania there is a complex referral system in which care must be stepwise (Fig 1). The system is set up in an organized way to triage less severe cases which can be managed at outside health facilities. While this system has potential benefits such as avoiding transfer in mildly injured children, our findings and others suggest that delaying presentation to definitive care in children that need transfer may confer greater risk for mortality. A study by Botchey et al. in Kenya found that injury patients transferred from another hospital had greater odds of in-

hospital mortality than patients admitted directly to the hospital [6]. Our findings suggest that outside facility healthcare workers may not know when to refer severely injured patients, and that there may be unreliable processes in place to facilitate such transfers, thus impeding care. Interventions that aim to bypass this complex system and refer severely injured children to definitive care sooner are needed. One way this could be done is by educating healthcare workers on acute stabilization of injured children and when to refer patients directly to a referral hospital rather than another intermediary facility.

Other ways to strengthen the healthcare system for pediatric injury patients is to improve access to transportation and increase enrollment in the existing national health insurance program. In Tanzania, like most LMICs, there is not a formal emergency medical system. This makes prompt response to trauma a challenge, and likely contributes to preventable morbidity and mortality in injured children [32–34]. Studies have shown high mortality rates in African emergency departments that suggest high acuity and inadequate access to quality prehospital and hospital-based emergency care [35,36]. The mortality rate in our cohort (8.2%) is relatively high in comparison to other similar studies, with the lack of an emergency medical system a likely contributor. A similar study in Rwanda did not find Delay 2 to be a major barrier [30], likely due to the strong pre-hospital system in Rwanda. Thus, capacity strengthening in pre-hospital emergency services in Tanzania is a pressing need to ensure timely access to definitive care. Further, the system of requiring payment for service is also causing delays at every point in the care continuum. Similar to our findings, a Delphi process that examined conceptual barriers to injury care in LMICs showed that cost was a barrier in both seeking care and in receiving timely care [37]. Only 22.62% of our patient population was enrolled in national health insurance. Thus, there is a need for increased enrollment in the national health insurance program that would remove the requirement for payment for service and thus streamline care.

Finally, we need to continue to improve quality of care for pediatric injury patients in the hospital setting. Although we did not find significant delays in receiving definitive care at KCMC, a few barriers were mentioned including wait time and delays due to treatment cost. A systematic review assessing LMIC trauma systems and applying the Three Delays Model found that the majority of the studies assessed Delay 3, whereas studies assessing delays 2 and 1 were assessed less often [38]. As the majority of the published trauma literature in LMICs is facility-centric, our study is a rarity by reporting on all three delays. In contrast to our findings, two studies applying the Three Delays Model to care seeking in LMICs found that delays in receiving quality care were most prominent [16,30]. This difference is likely due to the fact that this delay is highly setting-specific. Nevertheless, continuing to improve the receipt of quality care at the hospital level is imperative.

## Limitations

Since this study was based at KCMC and enrolled children that presented for care at KCMC, it is possible that there is selection bias. We are likely missing some children who presented to outside facilities for care and never made it to KCMC due to the long pre-KCMC timelines that function to pre-select surviving patients. This could contribute to missing data in Delays 1 and 2, and it also has the potential to skew the mortality data lower if these children died before arriving at KCMC. In addition, we did not collect quantitative or qualitative data at the outside facility level so we are missing both outcomes data and perspectives of patients who were not transferred. Further research should be done at the outside facility level to understand both outcomes and barriers more completely. In addition, since IDIs were conducted at KCMC, there is potential for social desirability bias influencing the interview responses especially in examining Delay 3. This social desirability bias could have made caregivers less comfortable in

discussing delays in receiving care at KCMC, which could be the reason we did not identify as many themes in this delay. In addition, we defined Delay 3 as delays in reaching appropriate care at KCMC, but we did not capture the quality of care received. Simply reaching appropriate care at KCMC does not equal high quality of care, and thus this is a limitation. Further, we did not include a variable on injury severity from the registry which could help to benchmark care quality performance with other settings. Finally, IDIs were not performed with caregivers of patients who had immediate mortality at KCMC due to ethical and logistical concerns of approaching these caregivers to request an interview immediately after the death of their child.

## Conclusions

Using the Three Delays Model, we were able to identify factors throughout the care continuum that contribute to delays in receipt of care for injured children. The complex referral pathway, financial constraints, and transportation challenges emerged as significant delays. In order to minimize delays and improve outcomes for pediatric injury patients, we need to develop interventions that increase caregiver and healthcare worker education, streamline the current trauma healthcare system, and improve timely and quality of care for pediatric injury patients in the hospital setting.

## Supporting information

**S1 Checklist. Inclusivity in global research.** Additional information regarding the ethical, cultural, and scientific considerations specific to inclusivity in global research. (DOCX)

## Acknowledgments

The authors would like to acknowledge the pediatric injury patients and their caregivers who participated in our study.

## Author Contributions

**Conceptualization:** Elizabeth M. Keating, Francis Sakita, Blandina T. Mmbaga, Catherine A. Staton, Melissa H. Watt.

**Data curation:** Elizabeth M. Keating, Ismail Amiri, Getrude Nkini, Nora Fino, Bryan Young.

**Formal analysis:** Sharla Rent, Nora Fino, Bryan Young.

**Funding acquisition:** Elizabeth M. Keating.

**Investigation:** Elizabeth M. Keating, Ismail Amiri, Getrude Nkini.

**Methodology:** Elizabeth M. Keating, Blandina T. Mmbaga, Ismail Amiri, Getrude Nkini, Catherine A. Staton, Melissa H. Watt.

**Project administration:** Elizabeth M. Keating, Francis Sakita, Blandina T. Mmbaga, Catherine A. Staton, Melissa H. Watt.

**Resources:** Elizabeth M. Keating, Francis Sakita, Blandina T. Mmbaga, Catherine A. Staton.

**Software:** Elizabeth M. Keating, Sharla Rent, Nora Fino, Bryan Young.

**Supervision:** Elizabeth M. Keating, Francis Sakita, Blandina T. Mmbaga, Catherine A. Staton, Melissa H. Watt.

**Visualization:** Elizabeth M. Keating, Catherine A. Staton, Melissa H. Watt.

**Writing – original draft:** Elizabeth M. Keating.

**Writing – review & editing:** Francis Sakita, Blandina T. Mmbaga, Ismail Amiri, Getrude Nkini, Sharla Rent, Nora Fino, Bryan Young, Catherine A. Staton, Melissa H. Watt.

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
