## [Decision Letter · Decision Letter 0]

8 Apr 2022

PGPH-D-22-00260

Three delays model applied to pediatric injury care seeking in Northern Tanzania: a mixed methods study

Dear Dr. Keating,

Thank you for submitting your manuscript to PLOS Global Public Health. After careful consideration, we feel that it has merit but does not fully meet PLOS Global Public Health’s publication criteria as it currently stands. Therefore, we invite you to submit a revised version of the manuscript that addresses the points raised during the review process.

We look forward to receiving your revised manuscript.

Kind regards,

Kathryn Chu, MD

Academic Editor

Journal Requirements:

1. Please include a complete copy of PLOS’ questionnaire on inclusivity in global research in your revised manuscript. Our policy for research in this area aims to improve transparency in the reporting of research performed outside of researchers’ own country or community. The policy applies to researchers who have travelled to a different country to conduct research, research with Indigenous populations or their lands, and research on cultural artefacts. The questionnaire can also be requested at the journal’s discretion for any other submissions, even if these conditions are not met.  Please find more information on the policy and a link to download a blank copy of the questionnaire here: https://journals.plos.org/plosone/s/best-practices-in-research-reporting. Please upload a completed version of your questionnaire as Supporting Information when you resubmit your manuscript.

2. Your co-author, Francis Sakita -francis.sakita@gmail.com, has not confirmed authorship of the manuscript. We have resent them the authorship confirmation email; however please check that the above email address for them is correct and follow up personally to ensure they confirm.  

3. We do not publish any copyright or trademark symbols that usually accompany proprietary names, eg (R), (C), or TM  (e.g. next to drug or reagent names). Therefore please remove all instances of trademark/copyright symbols throughout the text, including (C) on page 8.

4. Please update the completed 'Competing Interests' statement, including any COIs declared by your co-authors. If you have no competing interests to declare, please state "The authors have declared that no competing interests exist". 

5. In the online submission form, you indicated that "Data are available at reasonable request by contacting the corresponding author at elizabeth.keating@hsc.utah.edu. We are not able to deposit our data in a public repository given the ethical concerns of sharing qualitative data transcripts.". All PLOS journals now require all data underlying the findings described in their manuscript to be freely available to other researchers, either 1. In a public repository, 2. Within the manuscript itself, or 3. Uploaded as supplementary information.

Additional Editor Comments (if provided):

Reviewers' comments:

Reviewer's Responses to Questions

**Comments to the Author**

1. Does this manuscript meet PLOS Global Public Health’s publication criteria? Is the manuscript technically sound, and do the data support the conclusions? The manuscript must describe methodologically and ethically rigorous research with conclusions that are appropriately drawn based on the data presented.

Reviewer #1: Partly

Reviewer #2: Partly

2. Has the statistical analysis been performed appropriately and rigorously?

Reviewer #1: No

Reviewer #2: I don't know

3. Have the authors made all data underlying the findings in their manuscript fully available (please refer to the Data Availability Statement at the start of the manuscript PDF file)?

Reviewer #1: No

Reviewer #2: Yes

4. Is the manuscript presented in an intelligible fashion and written in standard English?

Reviewer #1: Yes

Reviewer #2: Yes

5. Review Comments to the Author

Reviewer #1: The study reports analysis of extracted data from 1 year of trauma registry cases at KCMC, Moshi. The authors explored differences in extracted characteristics between those who died or survived to discharge using univariable analysis. It also reports qualitative analysis of 30 conveniently sampled interviews of guardians of injured children mapped to the Three Delays.

I congratulate the authors for addressing an important topic and advancing the application of the Three Delays Model for evaluating injury care health systems.

I have some comments which I hope are useful to improve the paper.

• The authors report this as a convergent design mixed methods paper. A key tenet of mixed methods research, to my understanding, is that integration of the data / findings occurs to provide understanding / interpretation greater than the sum of its parts. I am not clear how the trauma registry data findings have been integrated with the interview data. Similarly the authors report triangulation across data sources, which to my mind would mean both registry and interview data, but this could be much clearer how these data have triangulated.

• The variable extraction from the registry could be better reported to explicitly state what variables were extracted. Is the registry on REDCAP or certain variables selected from the registry inputted into REDCAP?

• I struggled to follow the approach to analysis of the qualitative data. I find this whole section confusing as to how the anaylsis was conducted and by whom at which stage and how the iterations occurred. A flow diagram could perhaps clarify. Further, who wrote the memo and how? Was this written by the team as a team based approach? Condensing an hour interview to 2 sides represents a significant initial analysis to filter / code the data and who did this is important to understand.

• I would like to understand how the convenience sample was performed. The authors report 30 hours of interviews over 12 months from 2 interviewers (possibly 15 hours each). Can you elaborate why it took so long and in what way the sample was convenient?

• Univariable analysis was performed. I was a little confused as to how the analysis generated 4 p values for the mechanism of injury categorical variable but only 1 for the other categorical variables such as transport type and initial facility. I wondered whether a multivariable logistic regression analysis might have better summarised the associations between death and the variables explored, whilst controlling for confounders.

• If you do chose to use univariable categorical analysis where sizes are n<5 I think fisher’s exact should be used instead of chi squared. And a correction for multiple hypothesis testing e.g. Bonferroni, would be appropriate)

• The authors have conceived of interfacility delays within the third delay. Other researchers have categorised this as within the third delay (DOI:https://doi.org/10.1016/S2214-109X(20)30067-X) You may wish to reflect on this categorisation in the discussion.

• I found the narratives selected to be helpful and powerful illustrations of the issues summarised.

• When studying / discussing delay 3 I think the authors could have a broader understanding of delay 3 also encompassing the quality / appropriateness of care. Only considering purely a time delay to ED assessment will not capture anything about the quality of this ED assessment and the process a of the care delivered and the outcome for the patient. This should at least be acknowledged in the limitations.

Minor points:

• Ref 13 makes the point that the time critical nature of trauma is variable and the importance of timeliness depends of the nature of injury. This nuance should perhaps be reflected if using this reference.

• Registry inclusion and exclusion criteria descriptions are repetitive (lines 114 – 117).

• Money was provided for transportation costs. Did the caregivers have to travel to conduct the interviews? this is not clear.

• The abbreviation PI should be in full at first usage.

• Figure 2 labelling should reflect that it is both survivors and deaths being demonstrated.

• The first line of the discussion refers to care seeking delays, which is delay 1. But I think you’re referring to applying the 3 delays model to analysing paediatric injury care in an LMIC?

• The authors use the term intermediary to refer to facilities other than KCMC. Presumably these other hospitals are not always intermediary but rather primary and secondary care that can manage many things independently but need to refer to tertiary care for some conditions. I'm not sure intermediary is the correct/most helpful term.

• The first discussion paragraph refers to interventions. But I think your findings are identifying barriers rather than the solutions.

• Line 436 – I don’t think education on primary prevention of injury will influence care seeking after injury has occurred.

• When discussing mortality rates (line 464) you may wish to reflect on the long pre KCMC timelines that function to pre select surviving patients who have actually made it to tertiary care.

• In 470 increased enrolment in health insurance is discussed. This reads like such a system exists in Tanzania but is not commonly used. Is such a system present or are you proposing one should be developed? Or does this national insurance mean a taxpayer funded health system free for all to access?

• In limitations you may wish to reflect on the lack of injury severity variable from the registry reported to benchmark care quality performance with elsewhere / other standards.

• I don’t think reference 37 is properly represented. This reference study reported which of the three delays studies in the published literature have focused on. The authors own study is a rarity focussing on all three delays. Most of the published trauma care literature in LMCs is facility centric. Reference 37 is not really in contrast to this study’s findings.

• Re line 494 - perhaps explain social desirability and how that would have specifically potentially affected your findings.

• In figure 1 the regional and district hospitals are separately shown in the flow diagram but jointly analysed in the timeline to care arrows below. Perhaps be consistent

Reviewer #2: Thank you for the opportunity to review this manuscript on the three delays model as applied to pediatric injury in Tanzania.

One of the biggest challenges with discussing delays is being very precise on what is defined as a delay. To have a "delay" you would need to have set standard expectations in which a patient/caretaker should seek, reach, or receive care and this would likely vary based on the injury and severity. In this study, delay is not specifically measured or defined. Rather, this study reviews important factors which can influence delays and captures data from caretakers on perceived delays.

To better understand the context, it would be helpful to have more detail on the study setting. For example, is there an ambulance or EMS service? Are there pediatric surgeons at KCMC? What other services or healthcare facilities are available in the region? is this the only regional hospital with pediatric services? Is there a burn center at KCMC? What is the catchment area for KCMC? Each of these factors may play into the referral patterns and therefore potential delays.

In the quantitative analysis, did you capture data on severity of injury?

Was there any association between the demographic characteristics and delays? Is injury mechanism or illness severity associated with delays?

Some of the columns in table 1 total more than 100%. For example, the mechanism of injury totals 134.8%(37.5+11+34.2+52.1).

The first paragraph on the quantitative results (page 13, lines 213-218) describing the delays should be in the methods section rather than results.

As the quantitative analysis was conducted on a convenience sample, how can you be sure that this is representative of the patient population? Were patients who died included in the quantitative analysis? What were the characteristics of the patients selected for IDI?

Some of the themes were identified in only a couple of examples (money to pay bills, wait time). Is this really a theme if only 2-3 patients had this experience?

Was there any data captured on actual times for each level of delay? It seems that this data is really on patient and caretaker perceived delays, rather than actual medical delays. For example, there are different time expectations in regards to management of an open fracture compared with a closed fracture. Without this level of detail, it may be difficult to accurately capture data on delays.

In the discussion, you state that "delaying presentation to definitive care in children that need transfer is associated with a higher mortality risk (page 24, line 447-448)." I'm not sure that you can make that conclusion in this study. You do not know the mortality associated with patients who were not transferred (and managed at the other sites). Also, patients who were transferred may have had higher injury severity and therefore other factors which contributed to mortality.

Similarly, later in the same paragraph, you state that "intermediate facility healthcare workers may not know when to refer severely injured patients (page 24, line 451)." Again, you don't show data on the patients managed at the intermediate facility and you don't show data on illness severity.

You comment on the need for increased enrollment in health insurance. What percentage of this study population had health insurance?

6. PLOS authors have the option to publish the peer review history of their article (what does this mean?). If published, this will include your full peer review and any attached files.

**Do you want your identity to be public for this peer review?** For information about this choice, including consent withdrawal, please see our Privacy Policy.

Reviewer #1: **Yes: **John Whitaker

Reviewer #2: No

---

## [Decision Letter · Decision Letter 1]

25 Jul 2022

Three delays model applied to pediatric injury care seeking in Northern Tanzania: a mixed methods study

PGPH-D-22-00260R1

Dear Dr. Keating,

We are pleased to inform you that your manuscript 'Three delays model applied to pediatric injury care seeking in Northern Tanzania: a mixed methods study' has been provisionally accepted for publication in PLOS Global Public Health.

Best regards,

Julia Robinson

Staff Editor

Reviewer Comments (if any, and for reference):

Reviewer's Responses to Questions

**Comments to the Author**

1. If the authors have adequately addressed your comments raised in a previous round of review and you feel that this manuscript is now acceptable for publication, you may indicate that here to bypass the “Comments to the Author” section, enter your conflict of interest statement in the “Confidential to Editor” section, and submit your "Accept" recommendation.

Reviewer #2: All comments have been addressed

2. Does this manuscript meet PLOS Global Public Health’s publication criteria? Is the manuscript technically sound, and do the data support the conclusions? The manuscript must describe methodologically and ethically rigorous research with conclusions that are appropriately drawn based on the data presented.

Reviewer #2: Yes

3. Has the statistical analysis been performed appropriately and rigorously?

Reviewer #2: Yes

4. Have the authors made all data underlying the findings in their manuscript fully available (please refer to the Data Availability Statement at the start of the manuscript PDF file)?

Reviewer #2: Yes

5. Is the manuscript presented in an intelligible fashion and written in standard English?

Reviewer #2: Yes

6. Review Comments to the Author

Reviewer #2: (No Response)

7. PLOS authors have the option to publish the peer review history of their article (what does this mean?). If published, this will include your full peer review and any attached files.

**Do you want your identity to be public for this peer review?** For information about this choice, including consent withdrawal, please see our Privacy Policy.

Reviewer #2: No
